# Measurement of Supply-and Demand-Side Endowment Effects and Analysis of Their Influencing Factors in Agricultural Land Transfer

**Hanying Zhang** [†], **Jiafen Li** [†], **Jinlong Shen and Jianfeng Song** *

College of Economics and Management, Northwest A & F University, Yangling 712100, China
* Correspondence: s_jf@nwsuaf.edu.cn
† These authors contributed equally to this work.

**Abstract:** For a long time, the transference of land-contracting management rights was hindered by the unwillingness of farmers and inefficient use of farmland. The endowment effect is prevalent for farmers and is the main reason for the inhibited flow of farmland. However, recent studies have evaluated the endowment effect by treating buyers and sellers as one subject, which cannot be applied to individual land transfer transactions. Therefore, this paper redefines the endowment effect of farmland management rights by introducing market price as a reference point to evaluate the level of the supply- and demand-side endowment effects. The supply-side endowment effect is the gap between sellers' willingness to accept and the market rent; the demand-side endowment effect is the gap between the market rent and buyers' willingness to pay. In the empirical study, two multiple regression models are designed to measure the respective factors affecting the supply and demand sides, employing farm household survey data in Shaanxi Province. The obtained results reveal that the agricultural land transfer in Shaanxi Province is at a normal proportional level, and the supply- and demand-side endowment effects in agricultural land transfer are prevalent. The dependence of people on goods and the substitutability of the goods significantly affect the endowment effect of supply-side farmers, while the perception of expected income, cost and risk impact the endowment effect of demand-side farmers. Based on this, some policy recommendations are proposed to offset the influence of the endowment effect, thus promoting the development of the farmland transfer market.

**Keywords:** agricultural land transfer; endowment effect; influencing factors; loss aversion





## 1. Introduction

Decentralized management agriculture is the mainstay in China [1]. Due to the large population, limited land and the little amount of land available for each household, the operation scale is small [2,3]. However, small-scale, fragmented, decentralized, and fine-grained farmland operations have great negative impacts on agricultural production [4]. After the land reform, it became the basic orientation of agricultural policy to promote the transformation of agricultural operation methods by focusing on agricultural land transfer [5] and in 2013, the No. 1 central document Central Committee formally proposed to complete the registration and certification of rural land-contracting rights nationwide in five years. However, even after the confirmation of agricultural land rights, the pattern of fragmented agricultural operation has not undergone fundamental changes [6]. The transfer of contractual land management rights has encountered problems such as vague ownership of cultivated farmland, farmers' unwillingness to transfer and low efficiency in farmland utilization [7]. The development of China's farmland transfer market is still lagging behind. Scholars have found that the endowment effect universally existing among farmers is the main reason for hindering agricultural land transfer.

Endowment effects are found and explored in two main paradigms. In the barter paradigm, a subject is randomly assigned to an item the owner is reluctant to exchange [8]. In the valuation paradigm, a subject is randomly assigned to be either a buyer or a seller, suggesting that the minimum price for a seller who is willing to accept to sell an item (willingness to accept, WTA) is significantly higher than the maximum price that a buyer is willing to pay for the item (willingness to pay, WTP) [9–11]. The "endowment effect" phenomenon, that is, that WTA is always larger than WTP, was first proposed by Thaler in 1980 [12]. The endowment effect is defined as an individual's willingness to accept a good being more than they are willing to pay for it. The widespread endowment effects in economic goods challenge the economic rationality assumption. In addition, the endowment effect reveals a new behavioral economic mechanism: the expected price gap between trading parties leads to lower trading volumes, further affecting market efficiency. Currently, two main approaches measure the endowment effect; the impairment method (WTA minus WTP [10,13]) and the ratio method (WTA divided by WTP [14,15]). However, in both methods, WTA and WTP address the same subject. These methods conflate the supply and demand sides. Although they can better express endowment effects in the group, they do not facilitate individual transactions. When people give up a piece of land and acquire another one, they are influenced by moving/relocation costs, resulting in low rates of land transfer even without an endowment effect. However, the true endowment effect is due to the magnitude of the seller's psychological price above the market price and the buyer's psychological price below the market price, which inhibits transactions. Scholars have now raised questions about the traditional measurement method. Wang & Ou [16] set a neutral reference price in used-car pricing to verify the endowment effect. Individuals often base their decisions on reference points, and the difference between the outcome and the reference point can directly affect decision-making behavior. Anchoring refers to the influence of people in final value judgment as a reference point based on their initial exposure to a certain number [17].

Radin [18] found that the endowment effect is more severe for a property that is closely related to the personality of its holder than for substitutable property. In this context, agricultural land as a social welfare good provides basic survival security for villagers, making it a personified property for farmers. The endowment effect is more serious than that of fungible property [19]. Unlike countries with private ownership of land, there is no market for the sale and purchase of farmland ownership in China, but rather only a market for the transfer of farmland management rights, which is separate from the collective ownership of farmland and dependent on farmland contract rights [20]. For farmers who depend on agriculture for their livelihood, farmland is their main workplace and source of income. However, after the transfer of farmland, farmers lose the rights attached to the land and thus cannot enjoy the direct benefits that the land brings to them, which in turn causes the endowment effect. The endowment effect significantly weakens farmers' willingness to circulate land, thus distorting land market transaction prices in the process of land marketization. In addition, the endowment effect introduces the perspective of behavioral economics, which pays more attention to the micro-psychological characteristics of individuals and the irrational factors that influence their behavior and personalities.

In the agricultural land transfer market, the willingness to accept (WTA) is accurately assessed by the minimum amount required to give up his contracted land, reflecting the farmer's overall evaluation of his contracted land. The WTP is accurately assessed by the maximum amount paid to take a lease on someone else's land, reflecting the farmer's economic evaluation of his production and management capacity and land use efficiency [15,17,21,22]. This paper focuses on the endowment effect of special goods of farmland contract management rights in an active attempt to study the endowment effect theoretically. Meanwhile, this study introduces the market price (MP) as a reference point by redefining the endowment effect in evaluating the endowment effect of farmland management rights, defining the gap between WTA and MP as the supply-side endowment effect and the gap between MP and WTP as demand-side endowment effect. This study

focuses on the endowment effect of agricultural land management rights, measuring the supply-side and demand-side endowment effects of agricultural land-contracting management rights using farm household survey data in Shaanxi Province, and focuses on the effects of psychological dependence, objective dependence, and substitutability on the endowment effect of supply-side farmers and the effects of subjective expectations, cost perception, and risk perception on the endowment effect of demand-side farmers. Finally, based on the empirical results, corresponding incentives are designed to offset the endowment effect, stimulate the flow of farmland and bring the endowment effect from theory into practice.

## 2. Literature Review

Research on agricultural land transfer and the endowment effect has been richly documented, focusing on the connotations and occurrence mechanisms and measuring the endowment effect and its influencing factors on agricultural land transfer. The details are described as follows.

### 2.1. The Occurrence Mechanism and Measurement of the Endowment Effect

Thaler [12] was the first to observe that individuals tend to ask for more money than they are willing to pay when they acquire an item and define it as the endowment effect. There are rich results elaborating the mechanism of the endowment effect on common goods based on the standard economic theory, including loss aversion theory [12], mere ownership effect theory [23] reference price theory [24], regret [25], bias [26], exchange surplus [27] and natural property right evolution theory [28,29].

There are two main approaches to measuring the endowment effect. One is the impairment method (WTA minus WTP) [13]: when the result is positive, the endowment effect exists; if it is zero or negative, the endowment effect does not exist. Another is the ratio method (WTA divided by WTP) [14], suggesting that when the result is greater than 1, the endowment effect exists, and if it is less than or equal to 1, the endowment effect does not exist.

### 2.2. Research on the Endowment Effect in Agricultural Land Transfer

Specifically for agricultural land transfers, Zhong & Luo [30] first verified the existence of an endowment effect in agricultural land transfer and its inhibitory effect on agricultural land transfer. Subsequently, Nash & Rosenthal [31] verified that there was an endowment effect in the evaluation of students' preferred dormitories after the housing lottery. A follow-up survey shows that long-term experience can significantly increase the magnitude of the endowment effect even when other factors are controlled. Luo [6,32] provided an in-depth consideration of the reform and selection of China's agricultural land transfer system based on the endowment effect found in his research, emphasizing that the production institutional structure of property rights should be emphasized while deepening the research of the transactional institutional structure of property rights. Hu et al. [20] applied the factors influencing the endowment effect in general goods transactions to the agricultural land transfer context in China, and constructed a theoretical analysis framework of the factors influencing the endowment effect in agricultural land transfer from the perspective of human-land dependency relationship and human-land rights relationship, and analyzed the inter-generational differences in the influential mechanism of the endowment effect. Yan et al. [33] explained how the clarity, integrity, and stability of land subjective ownership under the "separation of powers" system affect farmers' endowment effects. Liu et al. [34] verified the existence of the endowment effect by studying the emotional attachment, property status, and substitutability of rural homesteads.

Research shows that there are many factors affecting the endowment effect. Firstly, for farmers on the supply side (i.e., farmland renters), the subjective psychological dependence on land, including strong, long-term experiences [31], emotional cognition [20], and emotional attachment [34,35] is one of the causes of the endowment effect; secondly, the

objective dependence of "people and land", including social security functions [36,37], legal empowerment [30], etc., is a factor; thirdly, the substitutability of income sources, etc., also affects the endowment effect [8,9,34,37].

For demand-side farmers (i.e., farmland lessees), the causes of their endowment effects are as follows: first, emotions about expected returns, including perceptions of identity rights, perceptions of control [20] and control [35]; second, perceptions of costs, including perceptions of technology adoption [38], for the cost of infrastructure inputs [38], the human cost incurred by farming, i.e. the number of people working on the farm; third, the perception of risk, including the source of income [34], uncertainty [9,30,39–41], and property rights status [30,34].

*2.3. Literature Gap and Contribution of the Study*

In the existing research literature, the existence and inhibition of the endowment effect in agricultural land transfer have been initially answered, while the causes of the endowment effect in agricultural land transfer are analyzed at the theoretical level; however, further in-depth research is needed on the following aspects.

First, the research on the connotation and occurrence mechanism of the endowment effect on farmland management rights requires further investigation. The endowment effect on agricultural land is different from that of ordinary goods. In contrast, China has a special property rights system for agricultural land, the institutional arrangement for collective ownership of rural land, and family contracting, determining the separation of land ownership and management rights. However, farming land management rights depend on contracting rights available to farmers' memberships in collective organizations [32]. In addition, China has a special relationship between people and land, and land has a dual function as a production source and social security for farmers. Due to a special emotional value, farmers live on land with high dependence on the land. This study focuses on the endowment effect of agricultural land-contracting management rights and the connotation and occurrence mechanism of agricultural land-contracting management rights as a special good, revealing the inner mechanism of endowment effect occurrence and making a positive attempt to study the endowment effect theoretically.

Second, measuring the endowment effect of agricultural land management rights confuses buyers and sellers, which obscures the reason for inhibited land transfer. The current method of measurement of the endowment effect, including the farmland management rights, is based on the definition of the endowment effect by two methods, namely the impairment method and the ratio method. However, in either method, WTA and WTP target the same subject. Treating buyers and sellers as a group to measure the endowment effect of agricultural land management rights can better express in the group; however, it does not promote individual transactions. The seller's expected price above the market price and the buyer's expected price below the market price inhibit transactions. The supply- and demand-side endowment effects are the gaps between the market rent and the supply and demand-side expected rent. These are measured in the specific field of farmland management rights and separately analyze the influencing factors on both sides to bring the endowment effect from theory into practice.

Finally, means of eliminating the impact of endowment effect on farmland transfer and designing corresponding incentive policies must be studied urgently. The endowment effect explains many perplexing economic phenomena and behavioral interventions, since the endowment effect can promote market efficiency, which is significant for designing the relevant incentives [29]. Previous studies describe the institutional arrangement of property rights to eliminate the endowment effect and promote agricultural land transfer [32]; however, there is a lack of generalized research on policy design. Based on the endowment effect, subsidizing agricultural land transfer according to time, place, and demand is worth studying. In addition, designing corresponding incentives to offset the endowment effect and thus stimulate agricultural land transfer is also required [42,43].

## 3. Theoretical Framework and Hypotheses

### 3.1. Theoretical Analysis of Factors Affecting the Supply-Side Endowment Effect

Emotional attachment produces loss aversion in the process of emotional divestment when people part with personally significant objects [34,44]. The more the emotional attachment an individual has to a specific good, the stronger the loss they tend to feel when they lose it and the higher the psychological price will be in comparison to its actual value [35]. Agricultural land brings farmer income and fulfils the desire for livelihood security [6], which enables Chinese farmers to have a natural emotional attachment to the land. Since rural China is a typical acquaintance society, neighbors' behavior also affects farmers' psychological perceptions, and thus their socioeconomic behavior [45]. Thus, agricultural land lessees will assign incremental value to their land because of this sense of loss, which is related to how long a person has held the right to use the good. As the duration of their tenancy lengthens, farmers' emotional attachment grows stronger, thereby enhancing their perception of the value of farmland and the resulting endowment effect [31,35]. We therefore propose the following hypothesis:

**Ha1:** *The deeper the farmers' emotional attachment to the land, the stronger endowment effect on the supply side.*

Some scholars have found that different kinds of goods create different dependencies for people [20], which leads to different kinds of loss-avoidance psychology and different valuations of prices in transactions, thus affecting the endowment effect. Under the premise of the "separation of three rights" of land in China, the registration and confirmation of land ownership are completed nationwide [6], but in terms of land management rights, a large number of transfers are only oral agreements without any contracts or formal registration [32]. As a point of reference, the mere ownership effect can cause a person to change their perception of value simply because they own an item [46]. Protecting the clarity of farmers' land management rights can deepen farmers' psychological sense of belonging to their farmland, weakening WTA and thus reducing the endowment effect. At the same time, since the implementation of the subsidy policy, the state has increased direct subsidies to farmers year by year, hoping to promote farmland transfer while further stabilizing farmland production and increasing farmers' income to gradually solve the current problems of small-scale farming and fragmented land division in rural China [47,48]. The satisfaction of farmers with the level of subsidies affects the valuation of WTA by farmers, which in turn reduces the endowment effect. We therefore propose the following hypothesis:

**Ha2:** *The higher the objective dependence degree of farmers, the stronger the supply-side endowment effect.*

Studies on the endowment effect have also focused on the substitutability of goods. It has been proposed that the lack of substitutability is what causes the endowment effect [8,9,37]. Substitutability refers to the significance of the target function of the good for the individual. In other words, the harder it is for a commodity to be substituted, the more it supports the individual, resulting in endowment effects [34]. If a farmer's employment status is not dominated by agriculture, then farmland is less valuable to the farmer. Therefore, as farmers become more educated, they become more selective about their jobs, the substitutability of farmland rises and individuals do not choose to farm, which is a type of work with a lower marginal rate of return, and the endowment effect is diminished. The supporting role of farmland for villagers is also reflected in villagers' livelihoods. If farmers own urban housing, farmers will give up agricultural land as livelihood capital owing to transportation costs as well as time costs. In this case, the endowment effect on villagers is not strong. We therefore propose the following hypothesis:

**Ha3:** *The stronger the substitutability of agricultural land, the weaker the supply-side endowment effect.*

The theoretical framework of factors affecting the supply-side endowment effect is shown in Figure 1.

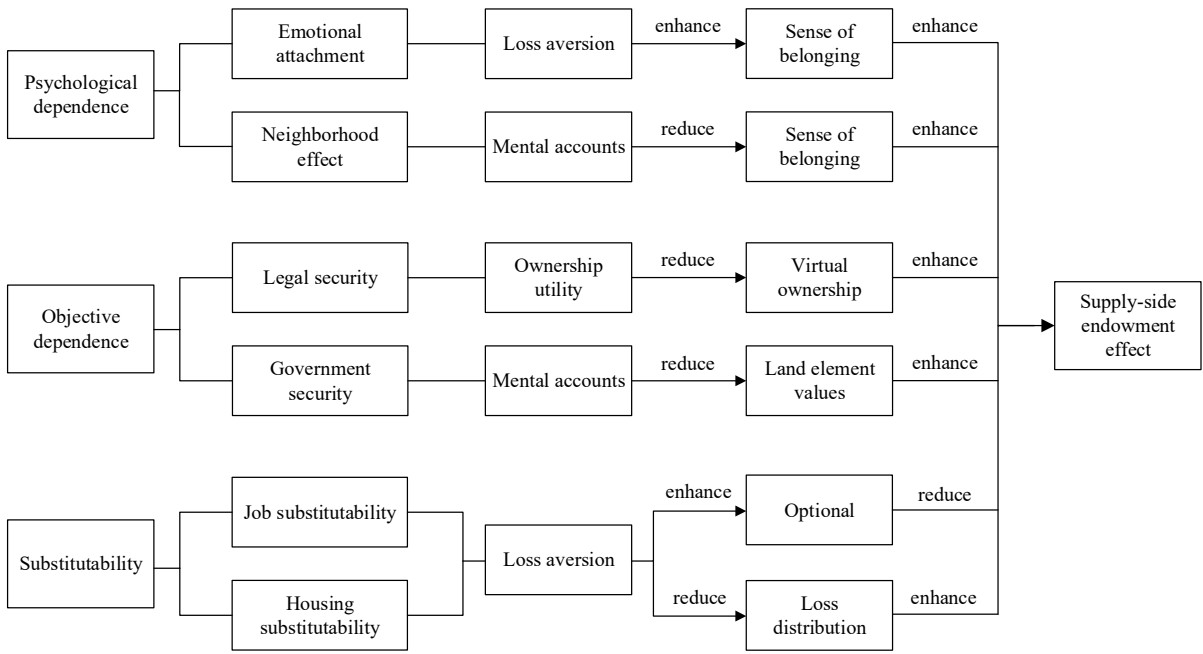

**Figure 1.** Influencing factors of supply-side endowment effect.

*3.2. Theoretical Analysis of Factors Affecting the Demand-Side Endowment Effect*

The endowment effect indicates that individuals have higher value judgments with regard to their items, and valuation is a highly subjective activity easily influenced by the valuator's emotions. Therefore, the endowment effect is also related to emotions. Recent research suggests that different emotions, such as positive, negative, and desired emotions, can influence the endowment effect [49,50]. However, the anticipatory negative emotions (especially regret and disappointment) prevent people from exchanging owned goods, enhancing the endowment effect [51]. Cosmides & Tooby [52] argued that there are specific psychological adaptations to social exchange, and the buyer's psychological expectations are among them. They create expectations based on the magnitude of returns relative to costs, causing differences in WTP [40]. For the demand-side farmers, the land is a production source, and they use moderate concentrations of land to develop large-scale operations and eventually to make profits. Therefore, the more demand-side farmers concerned about the land results in their higher subjective expectations of return on the farmland, the stronger the endowment effect. We therefore propose the following hypothesis:

**Hb1:** *The higher the farmers' subjective expectations of land, the weaker the demand-side endowment effect.*

Value construction serves as an approximate explanation for the endowment effect, where specific biases in sampling the attributes of potential trade items can lead to different value estimates for buyer and seller roles [28]. To assess the attractiveness of a transaction, decision-makers can compare the price to a reference price [24,53], and buying (selling) at a price above (below) the reference price is a "bad deal" [40]. It has been shown that exchange valuation incorporates the surplus that buyers and sellers seem to exchange [27]. Demand-side farmers cannot afford to have input costs greater than the possible benefits. The perception of cost inevitably enhances the endowment effect of the farmer, which in turn discourages the farmer's demand for farmland. The cultivation of farmland often requires more labor costs, time costs, and necessary pre-inputs, so this paper defines labor

inputs and infrastructure inputs into the cost perception dimension. and proposes the hypothesis:

**Hb2:** *The higher the perceived cost for farmers, the stronger the demand-side endowment effect.*

When the situation is full of uncertainty, individuals are more likely to experience loss aversion, and endowment effects are more likely to occur [8,21,40]. The uncertainty faced by farmers is diverse, and farmers' risk perceptions can be divided into two main categories. One of these is market risk perceptions. For demand-side farmers, if the uncertainty regarding the future cost of growing food increases, its optimal trading threshold is higher, resulting in less trading by the agency. The stability of property rights shapes the endowment effect mainly through individuals' emotional attachment to the good and to their subjective sense of ownership [54,55]. When a person has long-term possession of an item, the good is perceived as hard to replace [15], so the lessee of farmland is willing to reduce the WTP to offset some of the "loss aversion" of the lessor. The second is the perceived ability of the farm household. Non-farm employment risk refers mainly to the risk for farm households earning income from the non-farm sector [56]. As the demand-side farmers are among the main subjects of agricultural land transfer, the greater the non-farm employment risk, the greater the endowment effect, which in turn reduces their willingness to transfer. In this paper, two indicators, agricultural risk perception and property rights stability perception, are selected to incorporate market risk perception; farm and non-farm income are selected to measure farm households' ability perception. We therefore propose the following hypothesis:

**Hb3:** *The higher the risk perception of farmers, the stronger the demand-side endowment effect.*

The theoretical framework of factors affecting the demand-side endowment effect is shown in Figure 2.

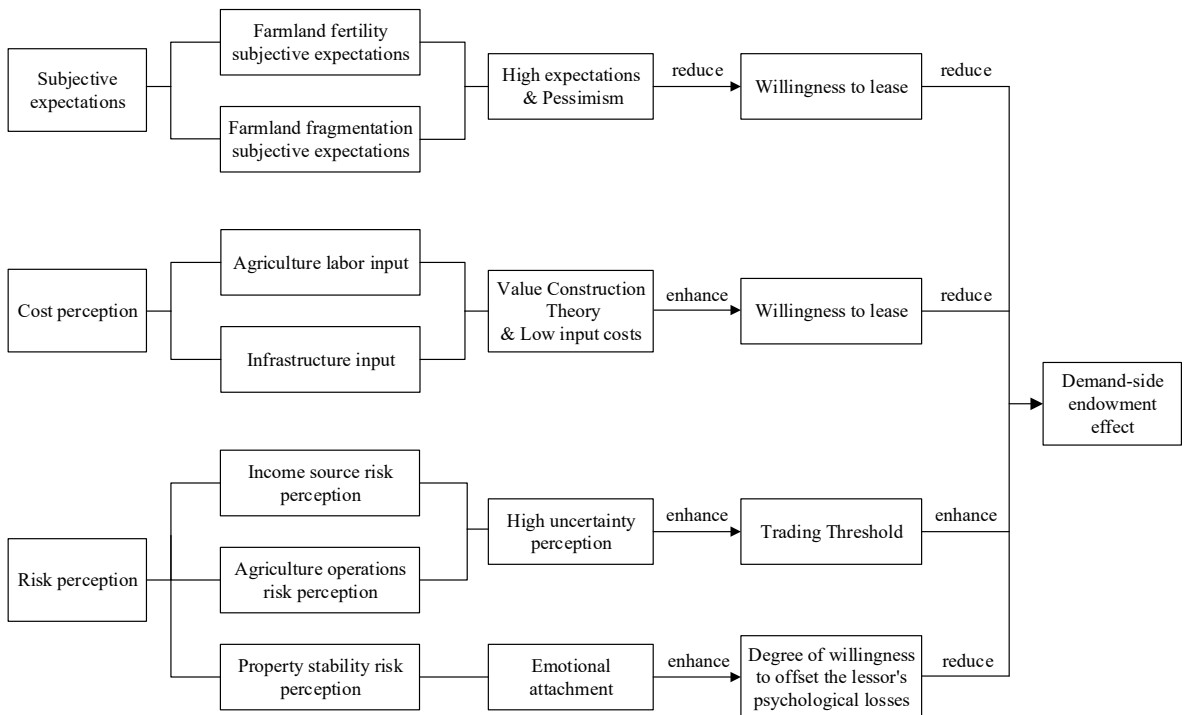

**Figure 2.** Influencing factors of demand-side endowment effect.

## 4. Research Methods and Data

### 4.1. Measurement of the Endowment Effect

The traditional measures of the endowment effect include the impairment method and the ratio method, while the ratio method can better express the existence and strength of the

endowment effect in the group but cannot measure the specific degree of loss in the seller's psychology. To concretely quantify the losses of farmers on both sides and to present the idea of a farmland transfer subsidy policy more effectively, this section uses the impairment method to calculate the supply-side endowment effect and the demand-side endowment effect. The conditional value assessment method is used, measured by interviews and experiments.

To validate and measure supply-side endowment effect, scenario assumptions were built into the questionnaire: (1) If you want to transfer your land, what is the minimum rent (WTA) you are willing to accept for one acre of land per year? (2) What is the rent at market price (MP) for one acre of land per year? The calculation formula is as follows:

$$\text{supply} - \text{side endowment} = \text{WTA} - \text{MP} \tag{1}$$

If the value of WTA -MP is greater than 0, the endowment effect exists; if the value of WTA -MP is equal to 0 or less than 0, the endowment effect does not exist.

To validate and measure demand-side endowment effect, scenario assumptions were built into the questionnaire: (1) What is the rent of market price (MP) for one acre of land per year? (2) If you want to expand the farmland, how much is the maximum rent (WTP) you are willingness to pay for one acre of land per year? The calculation formula is as follows:

$$\text{demand} - \text{side endowment} = \text{MP} - \text{WTP} \tag{2}$$

If the value of MP-WTP is greater than 0, the endowment effect exists; if the value of MP-WTP is equal to 0 or less than 0, the endowment effect does not exist.

*4.2. Models on Influencing Factors of the Endowment Effect*

Two multiple regression models were developed. Model 1 shows the effects of psychological dependence, objective dependence, and substitutability on the supply-side endowment of farmers. Model 2 shows the effects of subjective expectations, cost perception, and risk perception on the demand-side endowment effect of farmers. Two models were estimated with tolerance (TOL) values greater than 0.1 and variance inflation factor (VIF) values less than 10, indicating the absence of multicollinearity among the independent variables. Then, the regression analysis was conducted using Stata 15.1 software for the sample data investigated in the supply-side model and the demand-side model, respectively.

4.2.1. Models on Influencing Factors of the Supply-Side Endowment Effect

The quantified supply-side endowment effects obtained were continuous variables. In accordance with the characteristics of the variables and the purpose of the study, we used multiple linear regressions with multiple regression models to analyze the influencing factors of the endowment effect to verify the causal logic between the independent and dependent variables, and the regression results reflected the role and strength of the relevant influencing factors. The model is specified as follows:

$$y_i = \text{supply} - \text{side endowment effect} = \beta_0 + \sum_{i}^{n} \beta_i x_i + u_i \tag{3}$$

where the dependent variable $y_i$ is the "supply-side endowment effect value" of households; the independent variable $x_i$ is the vector of influencing factors affecting the size of the endowment effect which were selected according to the research hypotheses; $\beta_0$ is the constant term; $\beta_i$ is the coefficient of the independent variable, which indicates the degree and direction of the effect of a unit change in the independent variable on the dependent variable; $i$ is the independent variable number; and $u_i$ is the random error term. A detailed description of the variables is given in Table 1.

**Table 1.** Variable setting of supply-side endowment effects.

| Variable | Variable Descriptions | Note |
|---|---|---|
| **Dependent variable** | | |
| SEE | supply-side endowment effect | The difference between WTA and MP |
| **Independent variables: PD (psychological dependence)** | | |
| EA | emotional attachment | Planting years |
| NE | neighborhood effect | The number of households transferring out of the land within 500 m was summarized by GIS according to the latitude and longitude of the survey. |
| **Independent variables: OD (objective dependence)** | | |
| LS | legal security | Whether the contract should be signed: 1 = yes; 0 = no |
| GS | government security | What do you think about the level of agricultural subsidies?: 1 = low level; 2 = average; 3 = high level |
| **Independent variables: SUB (substitutability)** | | |
| JA | job substitutability | Education level: 1 = primary school and below; 2 = junior middle school; 3 = high school/technical secondary school; 4 = college; 5 = bachelor degree or above |
| HS | housing substitutability | Whether the household has urban housing: 1 = yes; 0 = no |
| **Independent variables: CV (Control variables)** | | |
| **IC (individual characteristic variables)** | | |
| AGE | age | Age of the household head (years) |
| GEN | gender | Gender of household head: male = 1, female = 0 |
| **FC (family characteristic variables)** | | |
| LH | low-income households | Whether they are low-income households: 1 = yes; 0 = no |
| CO | cooperatives | Whether they participate in cooperatives: "1 = yes; 0 = no" |
| | MDD (Market development degree variable) | |
| IO | intermediary organization | Availability of local circulation intermediary organization: 1 = yes; 0 = no |
| **FRE (farmland resource endowment characteristic variables)** | | |
| FA | farmland area | The actual owned farmland area of the household(mu) |
| LF | land fertility | The fertility of land: 1–5. The degree gradually increases from very bad/low to very good/high |

### 4.2.2. Models on Influencing Factors of the Demand-Side Endowment Effect

The quantified demand-side endowment effects obtained were continuous variables. In accordance with the characteristics of the variables and the purpose of the study, we used multiple linear regressions with multiple regression models to analyze the influencing factors of the endowment effect to verify the causal logic between the independent and dependent variables, and the regression results reflected the role and strength of the relevant influencing factors. The model is specified as follows:

$$y_j = \text{demand}-\text{side endowment effect} = \beta_0 + \sum_{j}^{n} \beta_j x_j + u_j \tag{4}$$

where the dependent variable $y_j$ is the "demand-side endowment effect value" of households; the independent variable $x_j$ is the vector of influencing factors affecting the size of the endowment effect, which were selected according to the research hypotheses; $\beta_0$ is the constant term; $\beta_i$ is the coefficient of the independent variable, which indicates the degree and direction of the effect of a unit change in the independent variable on the dependent variable; $i$ is the independent variable number; $u_i$ is the random error term. A detailed description of the variables is given in Table 2.

**Table 2.** Variable setting of the demand-side endowment effect.

| Variable | Variable Descriptions | Note |
|---|---|---|
| **Dependent variable** | | |
| DEE | demand-side endowment effect | The difference between MP and WTP |
| **Independent variables:SE(subjective expectations)** | | |
| SE1 | farmland fertility subjective expectations | Are you concerned about the fertility of farmland?: 1–5. The degree gradually increased, from very unconcerned to very concerned |
| SE2 | farmland fragmentation subjective expectations | Are you concerned about the farmland fragmentation?: 1–5. The degree gradually increased, from very unconcerned to very concerned |
| **Independent variables:CP (cost perception)** | | |
| ALI | agriculture labor input | Number of people working in agriculture in the household |
| II | infrastructure input | How much do they care about the condition of agricultural land infrastructure?: 1–5. The degree gradually increased, from very unconcerned to very concerned |
| **Independent variables:RP (risk perception)** | | |
| IS | income source risk perception | Annual gross household income for non-agricultural (yuan) |
| AO | agricultural operations risk perception | What is your overall perception of the stability of farmland property rights?: 1 = unstable; 2 = fair; 3 = very stable 4 = unclear |
| PS | property rights stability risk perception | Do you think the risk of farming operations is high?: 1 = very little; 2 = small; 3 = fair; 4 = large; 5 = very large |
| **Independent variables:CV (Control variables)** | | |
| **IC (individual characteristic variables)** | | |
| AGE | age | Age of the household head (years) |
| GEN | gender | Gender of household head: male = 1, female = 0 |
| **FC (family characteristic variables)** | | |
| LH | low-income households | Whether they are low-income households?: 1 = yes; 0 = no |
| CO | cooperatives | Whether they participate in cooperatives: "1 = yes; 0 = no" |
| **MDD (Market development degree variable)** | | |
| IO | intermediary organization | Availability of local circulation intermediary organization: 1 = yes; 0 = no |
| **FRE (farmland resource endowment characteristic variables)** | | |
| FA | farmland area | The actual owned farmland area of the household(mu) |
| LF | land fertility | The fertility of land: 1–5. The degree gradually increases from very bad/low to very good/high |

*4.3. Data Source and Sample Description*

Shaanxi Province is located in the gateway center of Northwest China, with a land area of 205,600 km2. By 2022, the permanent resident population was 39,550,000 in ten prefecture-level cities in the province. According to the data of China Statistical Yearbook, as a large agricultural province in China, Shaanxi Province has 3010.52 thousand hectares of arable land, which is a huge amount and accounts for a large proportion of the total agricultural land in China; and the number of agricultural population is also large, accounting for 40.57% in China. The land transfer in Shaanxi Province shows the characteristics of a wide geographical area, diversified subjects, and diverse forms, but its transfer procedure lacks standardized management and market-oriented mechanisms, which means that the sustainable growth of farmers' incomes cannot be guaranteed and there are serious and complicated problems. This situation is a serious challenge for the local government, and can be seen as a typical case of achieving scale in agricultural land management. Shaanxi agriculture is typically representative in China and even the world. Therefore, this study took Shaanxi Province as a case area with a view to providing a reference for efficient, market-oriented and standardized land transfer for the governance of similar regions, with implications for the theoretical analysis of the endowment effect and its application worldwide.

The data used in this study were obtained from the research conducted by the research team from September to October 2020. The research areas were Jingyang County Xianyang City, Weinan City Baishui County, Baoji City Fengxiang County, and Yijun County

Tongchuan City, Shaanxi Province, involving a total of thirteen villages in eight townships. These four counties constitute the Guanzhong area. Farmers studied were selected by proportional stratified random sampling. The survey was conducted in the form of questionnaires and interviews, and 35–40 farmers and 1–5 grassroots cadres were randomly selected from each village as research subjects on a household basis. The theme of the survey was "agricultural land transfer, endowment effect of agricultural land transfer, and agricultural scale operation", which involved relevant contents, such as agricultural land transfer, agricultural scale operation, and land property rights and endowment effect. The questionnaire contains basic information about individual farmers and their families (e.g., gender, age, years of farming, education level, family size, number of acres of farmland owned, land fertility income, etc.), land titling (e.g., land titling, views on titling, effects after titling), views on property rights (e.g., whether they agree that farmland is personally owned, general views on stability of property rights, etc. ), farmland policies (e.g., whether they participate in medical insurance and pension insurance, whether they have agricultural subsidies, etc.), the magnitude of relevant factors affecting the generation of endowment effects, and whether there is an endowment effect in the transfer of farmland by the respondents. A total of 458 valid questionnaires were obtained after excluding those with important information missing or contradictory data. The effectiveness rate of the questionnaire was 88.75%. The valid samples were distributed as 65 in Jingyang County, 141 in Baishui County, 132 in Yijun County, and 120 in Fengxiang County.

Among the respondents, 54.59% were male and 45.41% were female; 1.97% were 26–35 years old, 9.39% were 36–45 years old, 27.73% were 46–55 years old, 31.66% were 56–65 years old, and 29.26% were over 65 years old; 42.36% completed primary school education or below, 42.14% completed junior high school, 12.45% completed secondary school or high school, and 3.06% obtained undergraduate degrees or higher. As for family size, 24.67%, 69.21%, and 6.11% of the households had 1–3, 4–7, and more members, respectively. As for the situation of land circulation, 62.23% were self-farming farmers, 31.00% rented out the land, and 6.77% were farmers who operated both their land and land transferred from others.

The proportion of household agricultural income is the ratio of household agricultural income to total household income in a year, and it has a more dispersed distribution in quartiles. The average proportion of household agricultural income is 30.95%, which indicates that agricultural income no longer dominates farmers' incomes in the survey area. On the one hand, as a result of the advancements in poverty alleviation, a very large share of farmers' income is derived from subsidies to the agricultural industry; however, subsidy income is not included in agricultural income. Therefore, agricultural income in the field survey is limited to agricultural output and discounted cash income. On the other hand, this indicates that farmers' part-time employment is increasing. They continue to maintain semi-agricultural and semi-working status in the busy farming seasons. Among the sample households, 173 households or 37.7% had had the experience of transferring their land, which was about 25% fewer than those who did not have transfer experience. Thirty-six households transferred in the land, while 147 households transferred out the land; thus, the farmers who transferred out the land were much more numerous in comparison to those who transferred in the land. The latest data published by the Ministry of Agriculture reveal that by the end of 2020, nation-wide, 2582 counties (cities or districts) had carried out pilot projects, with a confirmed area of 850 million mu, or roughly 70% of the country's second-round contract area, and a national land transfer ratio of over 35%. By 2020, 39.7% of arable land in Shaanxi Province had been transferred, which was at the level of the normal transfer ratio.

## 5. Empirical Results and Analysis

*5.1. Measurement Results of the Endowment Effect*

5.1.1. Measurement Results of the Supply-Side Endowment Effect

According to the data, the local transfer rent varied due to the different degrees of development of the local agricultural land transfer market. Therefore, the farmers were divided into four categories according to geographical differences, and the data obtained from the survey were used to assign different market flow rents when measuring the endowment effect. The market rent in Fengxiang County was 700 yuan; in Yunmeng Township, Yijun County it was 750 yuan; in Yushuwan Village, Tai'an Town, Yijun County it was 550 yuan; in Jiao Ping Village, Tai'an Town, Yijun County it was 500 yuan; in Baishui County it was 450 yuan; and in Jingyang County it was 200 yuan.

The results of the supply-side endowment effect estimation (see Table 3) show that 46.50% of the farmers showed the endowment effect in the experiment on the valuation of land transfer rent; i.e., the minimum rent price they were willing to accept when transferring their agricultural land was higher than the local market rent price. Thus the prevalence of the supply-side endowment effect was verified in agricultural land transfer transactions, with an overall sample mean of 93.17.

**Table 3.** Statistical results of the supply-side endowment effect.

| Supply-Side Endowment Effect | Mean | Number of Samples (Household) | Proportion (%) |
|---|---|---|---|
| WTA-MP < 0 | −248.43 | 140 | 30.57 |
| WTA-MP = 0 | 0 | 105 | 22.93 |
| WTA-MP > 0 | 363.62 | 213 | 46.50 |
| total | 93.17 | 458 | 100.00 |

Note: 14 farmers did not rent and were treated as showing very high supply-side endowment effects, giving the maximum supply-side endowment effect.

However, 53.5% of farmers did not show the supply-side endowment effect (i.e., WTA-MP $\leqq$ 0) according to the survey interviews. There are two main reasons for this.

Firstly, some interviewed farmers held the opinion that the rental rates of land transfer were determined by market supply and demand, rather than personal will. The transferred land was primarily planted with food crops, and the net income of farmers' operations was already meagre, so if the price of the transferred land was too high, no one would take up the lease. In addition, in 2018 and 2019, some townships in the survey area were impacted by natural disasters and extreme weather, which reduced crop yields and dropped food prices, and significantly decreased farmers' willingness to transfer in land, with some villages even having no one to rent land. Second, in some villages, the arable land per capita was small and the distribution of plots was fragmented, so there were fewer renters in the land transfer market. The majority of the farmers who went out to work entrusted their land to their friends and relatives to use for farming, and only a small amount of rent was charged symbolically, or as a favour, no rent was charged. As a result, when transferring their land, some farmers were asked for the same or lower than market prices.

5.1.2. Measurement Results of the Demand-Side Endowment Effect

The results of the demand-side endowment effect estimation (see Table 4) show that 73.15% showed the endowment effect in the experiment on the valuation of land transfer rent; i.e., the local market rent price was higher than the maximum rent price they were willing to pay when transferring their agricultural land. Thus, the prevalence of the demand-side endowment effect was verified in agricultural land transfer transactions, with an overall sample mean of 260.56.

**Table 4.** Statistical results of the demand-side endowment effect.

| Demand-Side Endowment Effect | Mean | Number of Samples (Household) | Proportion (%) |
|---|---|---|---|
| MP-WTP < 0 | −159.77 | 64 | 13.97 |
| MP-WTP = 0 | 0 | 59 | 12.88 |
| MP-WTP > 0 | 386.75 | 335 | 73.15 |
| Total | 260.56 | 458 | 100.00 |

Note: 72 farmers did not lease farmland; it was considered that the farmer considered the farmland to be worthless and assigned WTP = 0.

However, 26.85% of individual farmers in the overall sample did not show the supply-side endowment effect (i.e., MP-WTP ≦ 0). According to the survey interviews, there were two main reasons for this:

First, some farmers who needed land and then leased it to engage in farming could anticipate reasonably high incomes in more efficient agricultural production activities, and large agricultural households could also expect to expand their large-scale operations and could expect relatively significant incomes. Second, because some farmers did not understand market supply and demand, market rent, and agricultural income, they were too optimistic about the expected income and were willing to pay relatively high rent or even more than the market rent.

### 5.2. Estimation of the Influence of the Endowment Effect
5.2.1. Estimation of the Influence of the Supply-Side Endowment Effect

As shown in Table 5, farmers' psychological dependence had a significant positive effect on the supply-side endowment effect. When farmers grew older, they had a stronger preference for "land love", "land cherishing", and "possession", forming a subjective illusion of value and overestimating the transaction price of agricultural land transfer. This suggests that the greater the emotional attachment to the land, the greater the psychological dependence of farmers on the land, and the stronger the endowment effect of agricultural land transfer.

The neighborhood effect of farmers had a significant negative effect on the supply-side endowment effect, and the regression coefficient of the neighborhood effect was significant at the statistical level of 1%. This paper defines a neighborhood as having a radius of 500 m, and verifies that the decision-making behavior of farmers is impacted not only by the farmers' characteristics, their family's characteristics, and their village's characteristics, but likewise by the decision-making behavior of their neighbors. This is due to the typical acquaintance societies seen in China's rural areas. Compared with urban area residents, farmers interact more frequently with one another and have closer neighborly relations. In this typical "relationship-based society", there is a mutual behavior learning among neighbors in agricultural land transfer. Therefore, the more the number of households in the 500 m neighbors transfer their land, the greater the willingness of farmers to transfer their land and the weaker the endowment effect in agricultural land transfer. Combining the above results, hypothesis a1 is verified.

Legal security had a significant negative effect on the strength of the supply-side endowment effect, which was significant at the statistical level of 1%. Both legal provisions and social acceptance support the signing of land contract management contracts (farmers' agreement rate for this is 91.05%, and the endowment effect of farmers who do not agree to sign the contract is more than four times that of those who agree to sign it). It can be seen that clarity of property rights facilitates the judgment of property rights transactions. This rule also applies to the transaction market of agricultural land use right transfer. Government security has a significant negative effect on the supply-side endowment effect. The regression was significant at the statistical level of 5%. In recent years, the central government has actively proposed strategies for rural social security and agricultural subsidy systems. Although the coverage of these systems is still limited and the level

of protection is still low, they reduce farmers' worries about uncertainty, suggesting that the clarity of land use rights brought about by contractual management contracts and the financial compensation from agricultural subsidies is conducive to diluting farmers' objective dependence on farmland, thus weakening its endowment effect and promoting farmland transfer. Combining the above results, hypothesis a2 is verified.

**Table 5.** Estimation results of the influence of the supply-side endowment effect based on the multiple regression model.

| Variable | Variable Descriptions | Coef. | *p*-value | S.E. |
|---|---|---|---|---|
| **Dependent variable** | | | | |
| SEE | supply-side endowment effect | | | |
| **Independent variables:PD(psychological dependence)** | | | | |
| EA | emotional attachment | 5.70 ** | 0.042 | 2.798 |
| NE | neighborhood effect | −17.35 *** | 0.007 | 6.412 |
| **Independent variables:OD(objective dependence)** | | | | |
| LS | legal security | −235.83 *** | 0.005 | 84.111 |
| GS | government security | −78.39 ** | 0.021 | 33.868 |
| **Independent variables: SUB(substitutability)** | | | | |
| JA | job substitutability | −53.73 * | 0.097 | 32.347 |
| HS | housing substitutability | −64.44 | 0.318 | 64.421 |
| **Independent variables: CV(control variables)** | | | | |
| **IC (individual characteristic variables)** | | | | |
| AGE | age | −6.66 * | 0.056 | 3.477 |
| GEN | gender | −10.13 | 0.843 | 51.206 |
| **FC (family characteristic variables)** | | | | |
| LH | low-income households | 173.26 ** | 0.023 | 75.838 |
| CO | cooperatives | −156.55 ** | 0.020 | 67.008 |
| **MDD (market development degree variable)** | | | | |
| IO | intermediary organization | 163.79 ** | 0.015 | 67.349 |
| **FRE (farmland resource endowment characteristic variables)** | | | | |
| FA | farmland area | −3.73 | 0.366 | 4.122 |
| LF | land fertility | −11.66 | 0.640 | 24.910 |
| Cons | | 858.20 *** | 0.000 | 198.643 |

N = 458
F(13, 444) = 3.70
Prob > F = 0.000
R-squared = 0.0977
Adj R-squared = 0.0713
Root MSE = 499.03

Note: *, **, *** represent 10%, 5%, 1% confidence level, respectively.

Job substitutability has a significant negative effect on the strength of the supply-side endowment effect. The explanation for this is that those disparities in literacy levels further contribute to the heterogeneity of the rural labor force by giving it different agricultural production capacities as well as non-farm employment capacities. Since the marginal returns of the agricultural sector in China today are significantly lower than those of the non-farm sectors, the more educated rural laborers are, the more they tend to allocate their labor to the higher-earning nonfarm sector. Thus, the higher the literacy level of farmers, the more jobs they have available and the lower the WTA, thus reducing the endowment effect of agricultural land.

Housing substitutability had a non-significant negative effect on the strength of the supply-side endowment effect, and did not pass the significance test at the 10% statistical level. There are a number of reasons for this. In the research, some farmers placed a high value on their land and farmland was still irreplaceable for these farmers. During the market economy transition, farmers' land complexity, though changing from thick to thin, has not been eliminated. Despite their wealth and higher income, having a piece of land in their hometown gave them a sense of security. Furthermore, with an hour's drive to the

county town, choosing to live in the city and driving to the countryside for farming has become possible. Combining the above results, hypothesis a3 is partially verified.

Among the control variables, age, whether they were low-income households, whether they participated in cooperatives, and whether there was a local intermediary organization for land transfer all affected the existence of the supply-side endowment effect in agricultural land transfer at the 5% or 10% significant level. Older farmers tended to have a greater love and appreciation for the land. They only transferred out of the land because physically they could not plant it. Thus, the supply-side endowment effect was weakened. Low-income households were more dependent on farmland, which made them give higher psychological value to farmland in transactions. Participation in cooperatives, on the other hand, reduced the probability of endowment effects in agricultural land transfer. Transfer intermediary organizations reinforce the supply-side endowment effect, since the current laws for land transfer intermediary organizations are inadequate and lack legalized constraints. Moreover, the rules for land use are uncertain due to the changes in interests and forces, which are highly unfavorable for intermediary activities. Moreover, with the intervention of land intermediaries, the unreasonable distribution of farmland transfer proceeds undermines farmers' entitled interests, negatively affects farmers' willingness and behavior toward land transfer, and slows down the farmland transfer. Gender and farmland resource endowment did not pass the significance test.

### 5.2.2. Estimation of the Influence of the Demand-Side Endowment Effect

As shown in Table 6, farmers' subjective expectations had a significant positive effect on the demand-side endowment effect. This is due to the fact that farmers have a strong sense of control over farmland, which suggests a high intensity of emotional connection to farmland, which consequently reinforces the psychological value of farmland. In contrast, individuals who care less about farmland are less reliant on the land, have a weaker attachment to the land, and show a stronger willingness to transfer their farmland when the benefits of switching to farmland are reasonable. Hence, the more farmers care about land fertility and land fragmentation, the more they overvalue the land, the more stringent their conditions for renting the land, and the more severe their negative expectations. Consequently, the endowment effect is exacerbated and discourages land transfer. Combining the above results, hypothesis b1 is verified.

Farmers' cost perception had a significant negative effect on the demand-side endowment effect, passing the test at the 1% statistical level. This is because the clearer the farmers' perception of the cost spent, the more favorable it is for farmers to transfer in the land. In terms of labor input, households with a high number of farmers tend to transfer in the land. It is not difficult to understand that farmers who transfer land do so for their reasons: they stay in their hometowns and rent land business, to increase the degree of utilization of labor resources and increase their income. Regarding infrastructure inputs, farmers consider the farmland's basic conditions when making decisions about contracting land. When the infrastructure conditions are fine, the demand-side farmers do not invest in infrastructure, and contracting the farmland is profitable, hence the endowment effect is relatively weaker. Combining the above results, hypothesis b2 is verified.

The perceived risk of income source and the perceived risk of property rights stability of farmland have significant negative effects on the demand-side endowment effect. A large number of rural people have moved to urban areas for work or self-employment, and their dependence on the land is weakening. When farmers' non-farm income is at a high level, they need less farmland to protect their income source, which reduces farmers' loss aversion in farmland transfer decisions. In terms of property rights stability, the agricultural land transfer is not simply a transaction of economic goods, but a transaction based on empowerment, emotion, and knowledge of rights and interests. The stability of land property rights is effectively related to farmers' land rights and interests, and farmers' subjective perception of land ownership will affect their valuation of the land. If a farmer

perceives that the property rights are relatively stable, he will feel more "secure" about the land and will be more willing to transfer in the land, which reduces the endowment effect.

**Table 6.** Estimation results of the influence of the demand-side endowment effect based on the multiple regression model.

| Variable | Variable Descriptions | Coef. | *p*-Value | S.E. |
|---|---|---|---|---|
| **Dependent variable** | | | | |
| DEE | demand-side endowment effect | | | |
| **Independent variables:SE(subjective expectations)** | | | | |
| SE1 | farmland fertility subjective expectations | 22.00 ** | 0.032 | 10.235 |
| SE2 | farmland fragmentation subjective expectations | 21.48 * | 0.078 | 12.153 |
| **Independent variables:CP (cost perception)** | | | | |
| ALI | agriculture labor input | −31.57 *** | 0.008 | 11.818 |
| II | infrastructure input | −34.45 *** | 0.002 | 10.797 |
| **Independent variables:RP (risk perception)** | | | | |
| IS | income source risk perception | −0.00 * | 0.099 | 0.000 |
| AO | agricultural operations risk perception | 39.41 *** | 0.001 | 12.276 |
| PS | property rights stability risk perception | −41.23 ** | 0.022 | 17.989 |
| **Independent variables:CV (Control variables)** | | | | |
| **IC (individual characteristic variables)** | | | | |
| AGE | age | 2.65 * | 0.021 | 1.146 |
| GEN | gender | −23.05 | 0.373 | 25.842 |
| **FC (family characteristic variables)** | | | | |
| LH | low-income households | 46.49 | 0.247 | 40.120 |
| CO | cooperatives | 68.09 * | 0.059 | 35.957 |
| **MDD (market development degree variable)** | | | | |
| IO | intermediary organization | 72.42 ** | 0.036 | 34.377 |
| **FRE (farmland resource endowment characteristic variables)** | | | | |
| FA | farmland area | −1.19 | 0.580 | 2.147 |
| LF | land fertility | 21.99 * | 0.096 | 13.198 |
| Cons | | 22.60 | 0.847 | 116.953 |
| N = 458 | | | | |
| F(14, 443) = 4.47 | | | | |
| Prob > F = 0.0000 | | | | |
| R-squared = 0.1238 | | | | |
| Adj R-squared = 0.0961 | | | | |
| Root MSE = 262.62 | | | | |

Note: *, **, *** represent 10%, 5%, 1% confidence level, respectively.

Farmers' perception of agricultural business risk has a significant positive effect on the supply-side endowment effect. The regression coefficient is 39.41 and is significant at the statistical level of 1%. Research revealed that the majority of farmers consider agricultural production a typical example of "living at the mercy of the elements", and a number of uncertainties will make farmers without sufficient financial capacity reluctant to transfer in land for cultivation. Moreover, farmers are already facing employment risks. As the cost of farming rises, they are more reluctant to farm if the land they transfer to does not generate income but exposes them to additional risks in farming. Therefore, the higher the farming operation risk perceived by the farmer, the stronger the degree of his endowment effect. Combining the above results, hypothesis b3 is tested.

Among the control variables, age among individual farmers' characteristics, whether they participate in cooperatives among farmers' household characteristics, the existence of local intermediaries for land transfer, and land fertility among farmland resource endowments all affect the existence of demand-side endowment effects in farmland transfer at the 5% or 10% significant level. The older the farmers are, the weaker their non-farm employment advantage and the more they rely on farmland operations for their survival, which reinforces the demand-side endowment effect. In contrast to supply-side farmers,

participation in cooperatives instead increases the probability of the occurrence of demand-side endowment effects in farmland transfer. The strength of the demand-side endowment effect is significantly enhanced by intermediaries, and the predicted benefits of engaging in farmland scale management have an impact on the transferee of farmland's engagement in the transfer. The reason for this is that with the intervention of intermediaries, farmers are hesitant to transfer their land because the meagre income they receive from cultivating the land does not amount to much after paying the intermediaries. When it comes to farmland resource endowment, the higher a farmer evaluates the fertility of the land he owns, the more likely he is to overestimate the price of farmland in farmland transfer transactions, and the more likely the endowment effect is to occur. Gender, whether a farmer was from a low-income household, and the number of acres of farmland owned in the farmland resource endowment did not pass the significance test.

## 6. Discussion

In this study, the supply-side and the demand-side endowment effects in the context of land-contracting management rights were redefined and measured, and an analysis of the influencing factors on the supply-side and demand-side endowment effects was carried out. As a result, we found that the endowment effect was prevalent on both the supply side and the demand side and was a significant factor in preventing the transfer of farmland. For the purpose of further promoting the development of the agricultural land transfer market, the psychological factors as well as the overall welfare level of farmers on both the supply side and demand side should be taken into account comprehensively. We can try to reduce the endowment effect of farmers in agricultural land transfer in the following ways.

For the transferring-out parties, first, in view of farmers' subjective feelings about land ownership, relevant institutions should make use of the opportunity of the confirmation of land-contracting management rights to further emphasize the significance of "three rights of separation" and correctly guide farmers' cognition of property rights in order to lessen the positive influence of farmers' psychological dependence on the endowment effect. Second, owing to the objective dependence of farmers on farmland, we should fully respect farmers' rights and interests in land transfer and make sure that they occupy the primary position in the land transaction while formulating the subsidy policy. It is farmers who should become the ultimate beneficiaries of land transfer. The transfer subsidies should be open, transparent, and precise to households. Simultaneously, the government should improve the social security mechanism for farmers leaving the land and minimize the risk factors for farmers after land transfer. Third, the citizenship of migrant workers faces numerous policy challenges due to the urban–rural binary system and other institutional reasons. These challenges include household registration, social security, and education of children who move with them, and the slow process of migrant workers' citizenship. Thus, we should accelerate the urban–rural integration reform of household registration, employment, medical care, education, and housing so as to promote the urbanization of migrant workers to increase the substitutability of agricultural land for farmers.

For the transferring-in parties, first, in the context of farmers' subjective expectations, the government can adjust tax policies to make tax incentives and financial support more friendly to agriculture. For example, the implementation of direct agricultural subsidies, seed selection subsidies, and agricultural machinery purchase subsidies can weaken the lessee's concerns about the condition of agricultural land, thus motivating agricultural land transfer. Second, in terms of cost perception, the township government can cultivate professional farmers with strong agricultural skills and rich management knowledge by organizing training courses and building practice bases to adopt the use of efficient agricultural land. More importantly, it is necessary to develop operators with better management capabilities to build a new type of agricultural business with higher efficiency and a larger scale of operation. On this basis, cooperation between professional farmers and large enterprises can promote the efficiency of agriculture by encouraging innovations and entrepreneurs. Finally, in response to farmers' perceptions of uncertain risks, the

government can improve infrastructure (e.g., transportation, water conservancy, energy, and information) and public services (e.g., education, medical care, social security, and pension services) in rural areas to improve the risk-prevention mechanism for farmers after land transfer.

Moreover, in the survey, it was found that 95% of the villages lacked a specialized flow organization and the presence of intermediaries increased the endowment effect. Therefore, governments should organize transfer agencies to popularize and guide reasonable expected market prices and further standardize transaction processes and contract forms. However, considering the positive effect of legal protection on the endowment effect, strict institutional management of transfer intermediaries is required, to prevent intermediaries from engaging in excessive intervention in the agricultural land market and becoming the dominant players in the land market. Therefore, to promote farmland transfer, it is equally necessary to increase government subsidies for participation in transfer as well as low government penalties for their casual default. In addition, rural land transfer is influenced by the productive costs incurred in the process of farmland management, which urges operators to actively learn high-tech technologies to reduce production costs while improving agricultural production efficiency.

Reducing farmers' endowment effects in farmland management rights is conducive to improving the efficiency of land transfer. Rural land transfer can provide farmers with land resources who have the ability as well as the willingness to expand their farming land. Moreover, it can provide income to farmers who would prefer to quit farming. Transferring agricultural land to capable and efficient farmers maximizes scale benefits, increases farmers' incomes, largely improves the efficiency of agricultural land usage, and enhances food security.

## 7. Conclusions

The fundamental purpose of agricultural land transfer is to improve the efficiency and economic benefits of land use in the process of rural urbanization and to develop scale operations, thus promoting agriculture modernization; however, there are endowment effects in real rural land transfer that limit its development. This paper measured and analyzed the factors affecting the endowment effect from both the supply side and demand side based on the survey data of farm households in the Guanzhong area in Shaanxi Province, and obtained the following main conclusions.

First, supply-side and demand-side endowment effects in rural land transfer were prevalent, with 46.50% of farmers having supply-side endowment effects and the mean value of the overall sample being 93.17. It was found that 73.15% of farmers had demand-side endowment effects, and the mean value of the overall sample was 260.56.

Second, the factors affecting the endowment effect of supply-side and demand-side farmers were different. For supply-side farmers, psychological dependence and objective dependence had positive and significant effects on the supply-side endowment effect, while substitutability had a significant negative effect on the supply-side endowment effect. For demand-side farmers, cost perception and risk perception had a significant positive effect on the demand-side endowment effect, while anticipated emotion had a significant negative effect on the demand-side endowment effect.

Third, with the increase in the farmer's age, the demand-side endowment effect significantly increased, while the supply-side endowment effect significantly decreased. Household characteristics had the same direction of influence on the supply side and demand side. Moreover, this study found that the presence of transfer intermediaries did not reduce the endowment effect of farmers to promote land transfer for both supply-side and demand-side farmers. On the contrary, it significantly increased the endowment effect and inhibited the farmland transfer.

This study focused on the endowment effect of agricultural land management rights, proposing a new concept and measurement method for supply-side and demand-side endowment effects, using agricultural land transfer market rent as a reference point. In

addition, it defined the difference between the potential demand-side psychological rent and the market rent as constituting the demand-side endowment effect and the difference between the potential supply-side psychological rent and the market rent as constituting the supply-side endowment effect. This concept accurately expresses why agricultural land flow is inhibited, providing a sufficient theoretical basis and data support for implementing agricultural land flow subsidy targets and subsidy amounts. Since the factors affecting the endowment effect of farmers on both sides are different, this study presented specific suggestions to offset the endowment effect owing to the target's identity. In addition, this study also attempted a theoretical study of the endowment effect, and proposed a policy idea for agricultural land transfer subsidies based on the research and empirical findings, providing a valuable reference for the government to formulate relevant policies. This concept has important practical significance for promoting agricultural land transfer and realizing a moderate-scale operation of agricultural production.

Reviewing the theories, methods, and conclusions of this study, there are still several shortcomings: first, theoretically, the mechanistic analysis of the factors influencing the endowment effect needs to be more rigorously deduced, the validity and reliability of the judgment of this paper need to be further confirmed, and the theory of endowment effect also needs a more scientific theoretical basis for further research and exploration. Second, the data analysis method of this study still adopted the two multiple regression models most commonly used for studying such problems. There is still room for improvement in the data analysis method and model result mining. In addition, with the advancement of marketization, farmers' dependence on land continues to weaken, and the endowment effect in farmers' agricultural land transfer is dynamically changing. Therefore, whether the existence of the endowment effect in farmers' agricultural land transfer still varies as time advances is also a direction worthy of consideration and research.

**Author Contributions:** Conceptualization, H.Z. and J.L.; methodology, H.Z.; software, H.Z.; validation, H.Z., J.L. and J.S. (Jinlong Shen); formal analysis, H.Z.; investigation, J.S. (Jianfeng Song); resources, J.S. (Jianfeng Song); data curation, J.S. (Jinlong Shen); writing—original draft preparation, H.Z.; writing—review and editing, H.Z. and J.L.; visualization, J.S. (Jinlong Shen); supervision, H.Z.; project administration, J.S. (Jianfeng Song); funding acquisition, J.S. (Jianfeng Song). All authors have read and agreed to the published version of the manuscript.

**Funding:** The Philosophy and Social Sciences Major Theoretical and Practical Issues Research Project of Shaanxi, Shaanxi, No.2021ND0380.

**Institutional Review Board Statement:** Formal ethics approval was not required by the funding body or the host academic institution.

**Informed Consent Statement:** Informed consent was obtained from all subjects involved in the study.

**Data Availability Statement:** The data from the interviews and surveys are deemed private.

**Conflicts of Interest:** The authors declare no conflict of interest.

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
