# Peer review of "Measurement of Supply-and Demand-Side Endowment Effects and Analysis of Their Influencing Factors in Agricultural Land Transfer"

_land, doi:10.3390/land11112053_

Round 1

Reviewer 1 Report

This study touched on some interesting ideas. However, I had a hard time understanding many aspects. Here are some more detailed comments and suggestions:

1.      I did not understand the third sentence of the abstract (even after reading the introduction). In what way do existing studies see buyers and sellers as a group? And how can this not reveal the essence of land transfer?

2.      I did not understand how the endowment effect for land is fundamentally different than the endowment effect for ordinary goods (e.g. as mentioned in the bottom paragraph on page 2). Could this be explained more clearly?

3.      I thought it was interesting to decompose the endowment effect into two parts, but it was not clear why this should be done based on the market price. In theory, what is the market price supposed to represent? Should WTA = MP and WTP = MP necessarily hold for someone who does not have an endowment effect? If not, then WTA > MP or WTP < MP (or both) could hold even without an endowment effect.

4.      The explanations of WTA and WTP at the beginning of Section 2.1 could probably be clearer and more precise.

5.      The definition of the endowment effect as “individuals are willing to pay more for a good than they are willing to receive it” is reversed.

6.      It might be useful to address moving/relocation costs. Couldn’t this help explain low rates of land transfer even without an endowment effect?

7.      I didn’t understand much of the discussion in the second-to-last paragraph on page 4.

8.      I didn’t really understand Hypothesis B1 and the discussion that comes right before it.

9.      The measures of WTA and WTP (as described in Section 3.1) seemed problematic. The measure of WTA is based on a question about the possibility of transferring one’s existing land – very crudely speaking, we can think of this as going from 1 unit of land to 0 units of land. Meanwhile, the measure of WTP is based on a question about the possibility of expanding one’s existing land – (again) very crudely speaking, we can think of this as going from 1 unit of land to 2 units of land. The issue then is that WTA and WTP (according to these measures) represent the value of two different things. WTA has to do with the value of the first unit of land while WTP has to do with the value of a second unit of land. This is an issue because standard economic theory would suggest that the value of a second unit of land would be less than the value of the first unit of land, due to diminishing returns. This suggests that, with these measures of WTA and WTP, we would expect measured WTP to be less than measured WTA **even without an endowment effect**. As a result, the measured gap between WTA and WTP may be capturing diminishing returns as opposed to capturing an endowment effect, and if so, it would call into question the interpretation of the analysis.

10.      In the measures of anticipated emotions (FF1 and FF2) seem inconsistent, subjects are asked “how much do they care,” while responses are coded on a 1-to-5 scale that ranges from “very unconcerned” to “very concerned.” However, one’s level of caring and one’s level of concern are two different things. For instance, someone can care a lot about farmland fertility without being concerned about it (e.g. if the farmer believes fertility is very important for output but is also optimistic regarding the fertility of the land).

Reviewer 2 Report

This paper analyzes the connotation of the endowment effect of agricultural land management, constructs the occurrence mechanism model of the endowment effect of agricultural land management by using farm household survey data in Shaanxi Province. The supply and demand-side endowment effects are defined as the gap between the market rent and the supply and demand-side expected rent. In the empirical study, two OLS regressions are constructed to measure the influencing factors on the supply-side and demand-side respectively. The results found that the agricultural land transfer in Shaanxi Province is at a normal proportional level, the supply-side and demand-side endowment effects in agricultural land transfer are prevalent, and the factors affecting the endowment effects of farmers on the supply-side and demand-side are different. The research topic attracts readers, but there are also the following points that can be improved:

1. Accuracy: The format of references is incorrect, such as the seventh line on page 4 “…… Morrison G.C., 1998”. In addition, please check that the format of references should be uniform, such as “Alchian, A. A. (1979). Some implications of recognition of property right transactions costs. In Economics Social Institutions (pp. 233-254). Springer, Dordrecht.”.

2. Literature review: There is no specific part of the literature review. The literature review mentioned in the introduction is too simple. Please supplement and summarize relevant literature research as much as possible. Make the structure of the paper more complete.

3. The variable names in Table 5 are suggested to use full names instead of abbreviations, so as to keep consistent with the full text table format.

4. The article mainly analyzes the influencing factors of the endowment effect from the supply side and the demand side. The empirical analysis of the influencing factors is too messy, and it is suggested to start from a specific point.

5. This article explores the influencing factors of endowment effect. Although the article is detailed and rich, what are the practical problems the article is committed to solving? The practical significance of this paper is not prominent enough.

6. Compared with previous studies, what are the innovations of this article? It is recommended to elaborate in the article.

7. The research is based on the survey data of farmers in Shaanxi Province, China. As an international journal, is the conclusion generally applicable to the world?

8. Finally, I have some suggestions which can improve the quality of the paper. Add recent (year-2021, 2022) literature references to strengthen the present work.

Reviewer 3 Report

The study focused on the endowment effect of agricultural land contracting management rights and has novelty in its measurement. Authors found demand side and supply side endowment effects which was influence by different factors. Thus, the study proposes offset the influence of endowment effect, thus promoting the development of the agricultural land transfer market. However, I strongly feel that the author must address the following serious issues in the paper to make it more appropriate for publication in the journal.

The abstract of the manuscript is well written and only one observation replace the keyword “emotional attachment” with “Factors”. The materials and methods used to arrive at the inferences are adequate and sufficient in the present study. Various methods used to estimate the factors affecting demand side and supply side endowment are available and the most appropriate among them were used in the present study. One suggestion to the authors, replace the term “OLS regressions” with “multiple linear regression models” in the methodology or in the abstract. Because ordinary least square (OLS) is the estimation method but, you use multiple linear models to draw your results.

The results of study are very well presented in tabular form in an organized way and are sufficient for academic needs of the present study. The inferences drawn from results of the study are well correlated with the available literature. The researcher has shown well acquaintance with the available literature pertaining to the present study. However, a few minor errors were noticed and marked in the manuscript. Keep the font of table 1, 2, 5 and 6 to times new romen. Provide the full form of “TOL” and “VIF” in the results. Keep the value of coefficient upto 2 digit after decimal in the text.

Overall, the researcher has produced a fine piece of research in terms of work done, neat presentation and proper contextualization of the results and it gives me pleasure to recommend for publication after minor revision.

Round 2

Reviewer 1 Report

.

Reviewer 2 Report

1. As an international journal, it is suggested to supplement the research significance of this manuscript for other countries around the world.

2.  The paragraphs in the discussion are too long. It is suggested to divide the paragraphs reasonably for description.

3.  Format problem: Please carefully check the format of the manuscript again, for example “……. differences in WTP (Drouvelis, M., & Sonnemans, J, 2017).”
